

# Extremely low levels of chloroplast genome sequence variability in *Astelia pumila* (Asteliaceae, Asparagales)

Simon Pfanzelt[1,2], Dirk C. Albach[2] and K. Bernhard von Hagen[2]

[1] Experimental Taxonomy, Leibniz Institute of Plant Genetics and Crop Plant Research, Gatersleben/Seeland, Saxony-Anhalt, Germany
[2] Institute of Biology and Environmental Sciences, Carl von Ossietzky University, Oldenburg, Lower Saxony, Germany

## ABSTRACT

*Astelia pumila* (G.Forst.) Gaudich. (Asteliaceae, Asparagales) is a major element of West Patagonian cushion peat bog vegetation. With the aim to identify appropriate chloroplast markers for the use in a phylogeographic study, the complete chloroplast genomes of five *A. pumila* accessions from almost the entire geographical range of the species were assembled and screened for variable positions. The chloroplast genome sequence was obtained via a mapping approach, using *Eustrephus latifolius* (Asparagaceae) as a reference. The chloroplast genome of *A. pumila* varies in length from 158,215 bp to 158,221 bp, containing a large single copy region of 85,981–85,983 bp, a small single copy region of 18,182–18,186 bp and two inverted repeats of 27,026 bp. Genome annotation predicted a total of 113 genes, including 30 tRNA and four rRNA genes. Sequence comparisons revealed a very low degree of intraspecific genetic variability, as only 37 variable sites (18 indels, 18 single nucleotide polymorphisms, one 3-bp mutation)—most of them autapomorphies—were found among the five assembled chloroplast genomes. A Maximum Likelihood analysis, based on whole chloroplast genome sequences of several Asparagales accessions representing six of the currently recognized 14 families (*sensu* APG IV), confirmed the phylogenetic position of *A. pumila*. The chloroplast genome of *A. pumila* is the first to be reported for a member of the astelioid clade (14 genera with c. 215 species), a basally branching group within Asparagales.

## INTRODUCTION

*Astelia pumila* (G.Forst.) Gaudich. is a dioecious, cushion-forming perennial herb. It is one of the main constituents of so-called Magellanic moorland (*Godley, 1960*), which prevails in the hyperoceanic fjord and channel landscape of West and Fuegian Patagonia of southwestern South America (*Schmithüsen, 1956*). The species occurs from 40°S to Cape Horn at 56°S, and on the Falkland Islands (Islas Malvinas). In the northern part of its range, it is found on the highest summits of the Chilean Cordillera de la Costa, which harbour isolated cushion peat bog outposts. Similar moorland enclaves occur also in the Northwest

Corresponding author
K. Bernhard von Hagen,
bernhard.vonhagen@uni-oldenburg.de

Patagonian Andes (*Heusser, Heusser & Hauser, 1992*; *Villagrán et al., 1998*; *Pfanzelt, García & Marticorena, 2013*). South of 47°S, the zonal vegetation is composed of cool-temperate *Nothofagus* rainforest and cushion peat bogs, where *A. pumila* is very abundant (*Pisano, 1983*; *Gajardo, 1994*). East of the Andes, in the arid Patagonian steppe, it is too dry for cushion peat bog development. *Astelia pumila* is probably insect-pollinated, however, flower visitors have never been observed during our own fieldwork. Its yellow berries were assumed to be bird-dispersed (*Skottsberg, 1905*). The species is probably tetraploid ($2n = 64$; *Moore, 1983*), with flow cytometric evidence that some individuals are hexaploid (S Pfanzelt, 2013, unpublished data).

*Astelia pumila* belongs to Asteliaceae, a small-sized family with three genera and 36 species from the circum-Pacific region, with most species occurring in the Southern Hemisphere (*Birch, 2015*). They grow in a variety of habitats, i.e., in forests, alpine fellfields and wetlands (*Bayer, Appel & Rudall, 1998*). The infrafamilial phylogenetic relationships were established by *Birch, Keeley & Morden (2012)*, based on DNA sequence data from chloroplast (*trnL*, *psbA-trnH*, *rps16* and *petL-psbE*) and nuclear (*NIA-i3*) regions. *Birch (2015)* revised the infrageneric classification of *Astelia*, but the placement of sect. *Micrastelia*, containing only *A. pumila*, remained unresolved (*incertae sedis*).

Together with other dominant cushion peat bog plant species, *Astelia pumila* is being used as a study system for the reconstruction of the ice-age history of Magellanic moorland with phylogeographic methods (*Pfanzelt, Albach & von Hagen, 2017*; *Šarhanová et al., 2018*). Previous genomic resources of *A. pumila* did not exist and our preliminary search for variable chloroplast markers did not produce satisfactory results. Consequently, the chloroplast genomes of five *A. pumila* individuals, sampled from almost the entire distribution range of the species, were assembled and compared, with the aim to identify phylogeographically informative chloroplast regions.

Here, the complete chloroplast genome sequence of *A. pumila* is reported and its intraspecific sequence variability assessed. Until now, there was no complete chloroplast genome sequence available of lower Asparagales, neither of Asteliaceae, nor of further astelioid genera (Boryaceae, Blanfordiaceae, Lanariaceae and Hypoxidaceae). Research on chloroplast genome evolution in Asparagales has been primarily focused on orchids (e.g., *Kim & Chase, 2017*; *Lin et al., 2017*; *Roma et al., 2018*) and Asparagaceae (e.g., *Steele et al., 2012*; *McKain et al., 2016*; *Floden & Schilling, 2018*). Major structural rearrangements have been documented in the chloroplast genome of parasitic and mycoheterotrophic species (e.g., *Barrett et al., 2014*), but in photoautotrophic members of the order, deviations from the typical land plant chloroplast genome structure are restricted to the loss of single genes (*Meerow, 2010*; *McKain et al., 2016*) and slightly shifting single copy-inverted repeat boundaries (*Dong et al., 2018*). Therefore, we did not expect the chloroplast genome of *A. pumila* to show large structural changes. However, the sequence data presented here may prove helpful to enhance our understanding of the evolutionary dynamics of the monocot plastome, through narrowing the sampling gap between orchids on the one hand and higher Asparagales on the other hand.

**Table 1  Information on the sequenced *A. pumila* specimens, respective DNA library types and collection localities.**

| Accession | Library type | Voucher | Collection locality | Geographic coordinates | GenBank accession number |
|---|---|---|---|---|---|
| ACMO.8 | whole gDNA | Pfanzelt 756 (CONC 180089) | Chile, Los Lagos, Villa Santa Lucía, Cuesta Moraga | 43.326°S, 72.390°W | MH752984 |
| AEX.3 | whole gDNA | Pfanzelt 477 (OLD) | Chile, Magallanes, Estero Excelsior | 52.554°S, 72.877°W | MH752983 |
| AFLK.3 | whole gDNA | Stanworth & Davey s.n. (OLD) | Falkland Islands, East Falkland | 51.680°S, 57.937°W | MH752980 |
| ALM | cp-enriched, cDNA | Pfanzelt 903 (OLD) | Chile, Los Lagos, Cordillera Sarao | 40.954°S, 73.731°W | MH752981 |
| AQU8.1 | whole gDNA | Pfanzelt & García Lino 535 (OLD) | Chile, Aysén, Queulat | 44.601°S, 72.439°W | MH752982 |

# MATERIALS & METHODS

## Sampling

As a non-model organism, for which genomic resources did not exist previously, next-generation sequencing was used to obtain DNA sequence data of five *A. pumila* individuals. Accessions were obtained from almost the entire distribution range of the species, including the Falkland Islands (Islas Malvinas), except for its northernmost occurrence at Cerro Mirador (40°S) of south-central Chile's Los Ríos Region.

## Illumina sequencing

Different library types were prepared: (1) a chloroplast-enriched library obtained via sorting on a BD FACSAria IIu cell sorter (using fresh leaf material; cf. *Wolf et al., 2005*) and subsequent whole genome amplification using the REPLI-g Mini Kit (Qiagen, Hilden, Germany), (2) whole genomic DNA libraries for shotgun-sequencing (based on silica-dried leaf material) and (3) a cDNA transcript library based on RNA extracted from fresh leaf material, using the RNeasy Mini Kit (Qiagen, Hilden, Germany). Libraries were paired-end sequenced on an Illumina HiSeq 2000 at the IPK Gatersleben (Germany), with an insert size of 400−500 bp. Information on the five sequenced *A. pumila* specimens, respective library types and collection localities is given in Table 1. Voucher specimens are deposited at the herbaria of the Universidad de Concepción, Chile (CONC), and Carl-von-Ossietzky-Universität, Oldenburg, Germany (OLD). The Chilean Corporación Nacional Forestal (18/2009) and the Falkland Islands Government (R10/2012) issued collection permits.

## Assembly of the chloroplast genomes

Removal of duplicate reads, adapter clipping and quality trimming was done in CLC Genomics Workbench (versions 6.5.1–7.5.1), setting the quality threshold to a qlimit of 0.001. To obtain a first chloroplast genome draft, the pooled quality-trimmed reads of all *A. pumila* individuals were mapped against *Eustrephus latifolius* (Asparagaceae, NCBI GenBank accession number KM233639.1) as a reference, using Geneious 8.0.5 (medium-low sensitivity and a five-time iteration; https://www.geneious.com). The resulting mapping was curated manually. Chloroplast contigs from de novo assemblies, performed in VelvetOptimizer 2.2.5. (*Zerbino & Birney, 2008*), were used to cross-check for

eventual mapping errors, especially of reads containing homopolymer stretches, and to fill missing regions. VelvetOptimizer hash lengths ranged from 19 to 63 and were optimized for N50 (optFuncKmer 'n50'). The chloroplast genome draft was then used itself as reference against which the reads of the individual *A. pumila* accessions were mapped, using Geneious 8.0.5 (five-time iteration, maximum 5% mismatches per read). The junctions between the large single copy (LSC) and the small single copy (SSC) regions and the two inverted repeats (IRs) were additionally validated through Sanger sequencing (LSC-IR$_B$ junction: Ap-rps19F AGACATGCGAGAAACGATAA, Ap-rps3R TGTGCGAACCAAAAGGAA; IR$_B$-SSC junction: Ap-IRbSSC-F CGAGTGAATGGAAAGGAAAA, Ap-IRbSSC-R TGGGGTTGGTGTTGTAAG; SSC-IR$_A$ junction: Ap-IRaSSC1F GGGGAGAAAGAAAG-GAAG, Ap-IRaSSC1R CGGGAATCATTAGGAAGT; IR$_A$-LSC junction: Ap-trnHF ATTCACAATCCACTGCCT, Ap-psbAR TGCTCACAACTTCCCTCT).

## Genome annotation

Chloroplast genome annotation was done using DOGMA (*Wyman, Jansen & Boore, 2004*; for reference chloroplast genomes, see http://dogma.ccbb.utexas.edu/html/cp_taxa), and cross-checked using GeSeq (*Tillich et al., 2017*) and the "Annotate from ..." function in Geneious. Via that latter function, annotations can be transferred from a user-specified reference set of chloroplast genomes to the *A. pumila* target. The chloroplast genome of *Asparagus officinalis* (GenBank accession number NC_034777.1) was used as reference when employing GeSeq and Geneious for genome annotations. Where necessary, gene boundaries were corrected manually to match start and stop codons. The annotated chloroplast genome sequences were submitted to GenBank (accession numbers MH752980–MH752984). Chloroplast genome maps were drawn using OGDRAW. Both OGDRAW and GeSeq are available at the MPI-MP CHLOROBOX website (https://chlorobox.mpimp-golm.mpg.de/index.html).

## Intraspecific sequence comparisons and phylogenetic reconstruction

The chloroplast genome sequences of the five *A. pumila* specimens (Table 1) were aligned with MAFFT (*Katoh et al., 2002*) and screened for variable sites. Coverage cutoff was set to 100 in order to retrieve reliable markers. A NeighborNet was constructed using SplitsTree 4.14.6 (*Huson & Bryant, 2006*) based on HKY85 distances. To confirm the placement of *A. pumila* within Asparagales, 22 chloroplast genome sequences, representing six of the 14 currently recognized families of the order (*sensu APG IV, 2016*), were downloaded from NCBI GenBank and aligned using MAFFT (*Katoh et al., 2002*), together with the chloroplast genome sequence of *A. pumila* individual AEX.3. There are no complete chloroplast genome sequences available yet of the remaining families of Asparagales (Boryaceae, Blandfordiaceae, Lanariaceae, Hypoxidaceae, Doryanthaceae, Ixioliriaceae, Tecophilaeaceae and Xeronemataceae). A phylogenetic tree was constructed using a Maximum Likelihood approach as implemented in RAxML 8.2.0 (*Stamatakis, 2014*). In a single run, a rapid bootstrap analysis and a best-scoring ML tree search were carried out, using the GTRGAMMA model of nucleotide substitution and 1,000 bootstrap replicates. *Alstroemeria aurea* (Liliales) served as outgroup.

## RESULTS

The total lengths of the individual *A. pumila* chloroplast genome sequences vary from 158,215 bp to 158,221 bp due to indel variation (Fig. 1). The large and small single copy regions have lengths of 85,981–85,983 bp and 18,182–18,186 bp, respectively. The inverted repeat regions have a length of 27,026 bp. GC content is 37.8%. Genome annotation predicted a total of 113 genes, including 30 protein-coding genes and four rRNA genes.

Intraspecific chloroplast sequence variation was very low in *A. pumila*. Over a length of 158 kb, 37 variable sites were found, of which 18 were indels, 18 single nucleotide polymorphisms (SNPs), and one a 3-bp mutation (Table 2). The latter occurred in an imperfect repetitive region and was treated as a single mutation event. All the SNPs and the 3-bp mutation were autapomorphies of which 10 occurred in non-coding regions, i.e., introns or spacers. Of the eight SNPs occurring in coding regions, four represented non-synonymous mutations. Read coverage at the SNP sites ranged from 139−2,484 (mean 468, s.d. 541). The 18 observed indels had lengths of 1−2 bp. Indel variation was always associated with homopolymer runs of maximally 15 bp length. No differences in homopolymer lengths were observed when cross-checking the Geneious mappings with the contigs of the VelvetOptimizer de novo assemblies, so indel variation was not a software-related artefact. The NeighborNet showed a star-like topology (not shown).

The phylogenetic reconstruction of Asparagales, based on whole chloroplast genome sequences, recovers Orchidaceae as basally branching within the order (Fig. 2). The astelioid clade, represented in this study by Asteliaceae, is then sister to the remaining Asparagales.

## DISCUSSION

The chloroplast genome of *A. pumila* showed the typical quadripartite structure, i.e., a large and a short single copy region and two inverted repeats (Fig. 1). No general differences in gene order or inversions were detected when comparing the *A. pumila* chloroplast genome to those of related Asparagales species. In general, major structural rearrangements like, for example, the IR enlargement and inversions documented for geranium (*Palmer, Nugent & Herbon, 1987*) or the 22-kbp inversion that marks an early evolutionary split in Asteraceae (*Jansen & Palmer, 1987*), have not been detected yet in the chloroplast genome of any Asparagales species. However, gene loss has been documented for some taxa throughout higher Asparagales (*Meerow, 2010*; *Steele et al., 2012*; *McKain et al., 2016*). These missing genes—*clpP*, *ndhF*, *rpl32*, *rps16*, and *rps19*—are all present in the chloroplast genome of *A. pumila*. In orchids, basally branching within Asparagales, degradation of the *ndh* gene complex has been frequently observed, especially among heterotrophic species (*Neyland & Urbatsch, 1996*; *Chang et al., 2006*; *Lin et al., 2017*). By contrast, all eleven *ndh* genes are maintained in *A. pumila*. *McKain et al. (2016)* identified the *rps19* gene to be the most dynamic in Agavoideae (Asparagaceae). There, it was either missing, pseudogenized, or present at different positions, either within the LSC or the IR. In *A. pumila*, *rps19* is found within the IR, close to the LSC-IR boundaries. Located between the *rps19* and the *psbA* genes, there is a partial *rpl22* gene, truncated at the LSC-IR$_A$ junction. This kind of gene order was classified as Type IIIg by *Wang et al. (2008)*, a configuration typically found

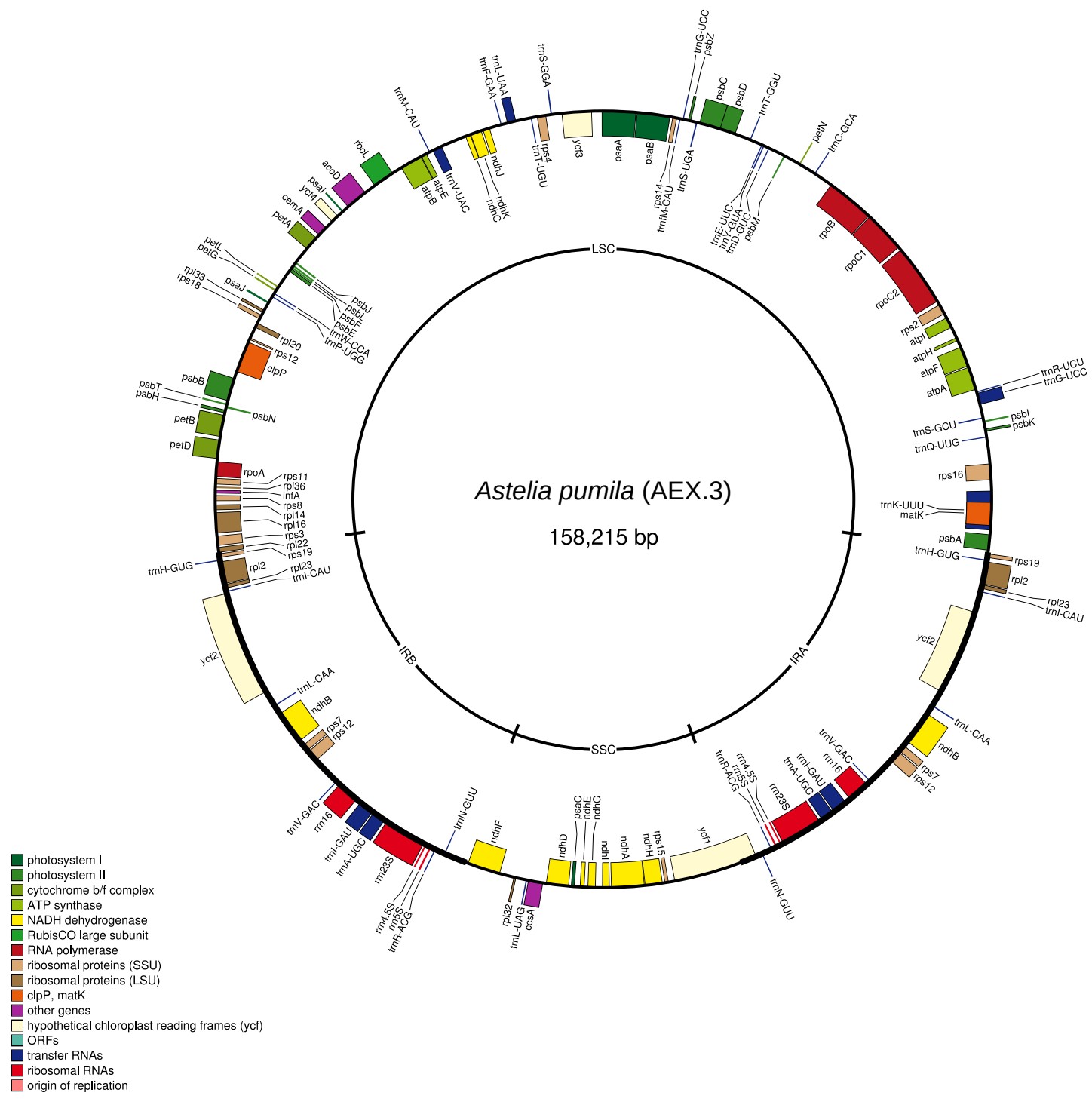

**Figure 1** **The chloroplast genome map of *Astelia pumila* specimen AEX.3 (see Table 1).** Genes shown on the outside of the outer circle are transcribed clockwise, while genes shown on the inside are transcribed counterclockwise. The positions of the large (LSC) and small single copy (SSC) regions, as well as of the inverted repeats (IRs), are indicated on the inner circle.

**Table 2** **Variable positions among five compared whole chloroplast genome sequences of *Astelia pumila*.** Indel variation was exclusively associated with homopolymer stretches, of which length and identity of the repeat unit are given.

| Position | Region | Type | ACMO.8 | AEX.3 | AFLK.3 | ALM | AQU8.1 | SNP category |
|---|---|---|---|---|---|---|---|---|
| 73432 | *clpP* intron 1 | 3 bp | AAA | AAA | AAA | AAA | TTT | n. a. |
| 9356 | *trnS-GCU-trnS-CGA* spacer | indel | 13 A | 15 A | 13 A | 13 A | 13 A | n. a. |
| 14799 | *atpI-atpH* spacer | indel | 12 T | 11 T | 12 T | 12 T | 12 T | n. a. |
| 23333 | *rpoC1* intron | indel | 10 T | 11 T | 10 T | 10 T | 10 T | n. a. |
| 29135 | *trnC-GCA-petN* spacer | indel | 10 A | 9 A | 10 A | 10 A | 10 A | n. a. |
| 37554 | *psbZ-trnG-UCC* spacer | indel | 11 A | 11 A | 11 A | 10 A | 11 A | n. a. |
| 46595 | *trnS-GGA-rps4* spacer | indel | 10 A | 10 A | 11 A | 10 A | 10 A | n. a. |
| 48875 | *trnL-UAA* intron | indel | 8 A | 8 A | 9 A | 8 A | 8 A | n. a. |
| 49336 | *trnL-UAA-trnF-GAA* spacer | indel | 12 T | 12 T | 12 T | 12 T | 13 T | n. a. |
| 57002 | *atpB-rbcL* | indel | 13 T | 13 T | 13 T | 13 T | 14 T | n. a. |
| 72632 | *clpP* intron 2 | indel | 12 T | 13 T | 12 T | 12 T | 12 T | n. a. |
| 73328 | *clpP* intron 1 | indel | 12 T | 12 T | 10 T | 13 T | 13 T | n. a. |
| 73666 | *clpP* intron 1 | indel | 13 T | 11 T | 12 T | 12 T | 12 T | n. a. |
| 83264 | *rpl14-rpl16* | indel | 13 T | 11 T | 11 T | 12 T | 11 T | n. a. |
| 84702 | *rpl16* intron | indel | 14 T | 14 T | 15 T | 14 T | 14 T | n. a. |
| 115430 | *ndhF-rpl32* spacer | indel | 10 A | 9 A | 10 A | 10 A | 10 A | n. a. |
| 116569 | *rpl32-trnL-UAG* spacer | indel | 13 T | 11 T | 13 T | 12 T | 12 T | n. a. |
| 116734 | *rpl32-trnL-UAG* spacer | indel | 14 T | 13 T | 13 T | 13 T | 13 T | n. a. |
| 121696 | *ndhG-ndhI* spacer | indel | 10 T | 10 T | 11 T | 10 T | 10 T | n. a. |
| 1662 | *trnK-UUU* intron | SNP | C | C | C | C | A | n. a. |
| 2873 | *matK* CDS | SNP | G | G | G | T | G | non-synonymous S →Y |
| 16456 | *rps2* CDS | SNP | G | G | G | A | G | synonymous |
| 18775 | *rpoC2* CDS | SNP | C | A | C | C | C | non-synonymous L →F |
| 31960 | *trnD-GUC-trnY-GUA* spacer | SNP | T | C | C | C | C | n. a. |
| 41582 | *psaA* CDS | SNP | T | C | T | T | T | synonymous |
| 43551 | *psaA-ycf3* spacer | SNP | C | T | C | C | C | n. a. |
| 50166 | *trnF-GAA-ndhJ* spacer | SNP | G | G | G | G | T | n. a. |
| 54434 | *trnC-ACA* intron | SNP | G | A | G | G | G | n. a. |
| 64527 | *petA* CDS | SNP | T | C | T | T | T | synonymous |
| 69323 | *psaJ-rpl33* | SNP | A | A | A | C | A | n. a. |
| 78402 | *petB* exon 2 | SNP | G | G | G | A | G | synonymous |
| 80834 | *rpoA* CDS | SNP | T | T | T | T | G | non-synonymous L →F |
| 113101 | *trnN-GUU-ndhF* spacer | SNP | A | A | T | A | A | n. a. |
| 116443 | *rpl32-trnL-UAG* spacer | SNP | C | C | C | A | C | n. a. |
| 116625 | *rpl32-trnL-UAG* spacer | SNP | G | G | G | G | T | n. a. |
| 118257 | *ccsA-ndhD* spacer | SNP | T | A | T | T | T | n. a. |
| 127013 | *ycf1* CDS | SNP | C | A | C | C | C | non-synonymous R →L |

**Notes.**

Amino acid codes: F, phenylanaline; L, leucine; R, arginine; S, serine; Y, tyrosine; n. a., not applicable.

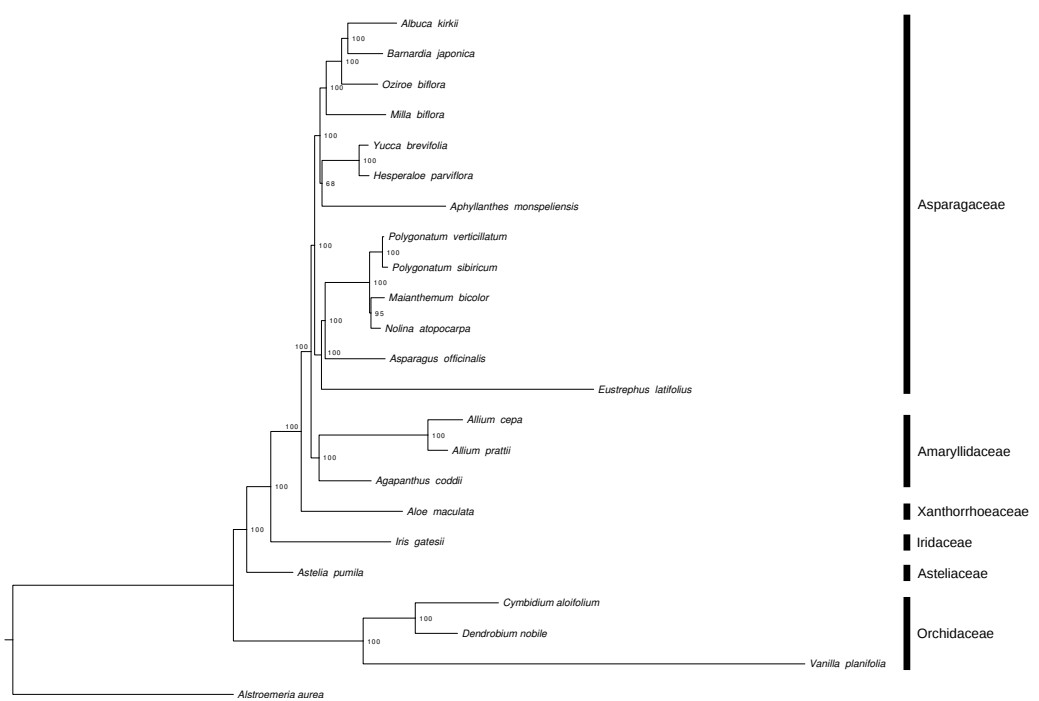

**Figure 2** **Maximum Likelihood tree, based on whole chloroplast genome sequences, to illustrate the phylogenetic position of *A. pumila* within Asparagales.** *Alstroemeria aurea* (Liliales, Alstroemericeae) served as outgroup. Numbers indicate node support (based on 1,000 bootstrap replicates). NCBI GenBank accession numbers: *A. aurea*, KC968976; *V. planifolia*, KJ566306; *D. nobile*, KX377961; *C. aloifolium*, NC_021429; *A. pumila*, MH752983; *I. gatesii*, NC_024936; *A. maculata*, KX377523; *A. coddii*, KX790363; *A. prattii*, MG739457; *A. cepa*, NC_024813; *E. latifolius*, NC_025305; *A. officinalis*, KY364194; *N. atopocarpa*, KX931462; *M. bicolor*, NC_035970; *P. sibiricum*, KT695605; *P. verticillatum*, KT722981; *A. monspeliensis*, KX790360; *H. parviflora*, NC_032703; *Y. brevifolia*, NC_032711; *M. biflora*, KX822778; *O. biflora*, NC_032709; *B. japonica*, KX822775; *A. kirkii*, NC_032697.

in Asparagales and Commelinales. In other Asparagales, e.g., *Asparagus densiflorus* and *Crinum asiaticum*, the LSC-IR$_A$ junction lies downstream of the *rps19* gene and the IR$_A$ does not include a partial *rpl22* gene. The structural dynamics of the LSC-IR junctions carry a phylogenetic signal, since there is an IR expansion trend in monocots: basally branching groups have generally shorter IRs than derived ones (*Wang et al., 2008*).

Eighteen indels of 1−2 bp length were observed among the five *A. pumila* accessions compared, all of which were associated with A or T homopolymer stretches of 8 to 15 bp length. It has been shown that the indel error rate of the Illumina sequencing platforms increases after long homopolymer runs by up to two orders of magnitude (*Ross et al., 2013*). Therefore, indel variation associated with homopolymer stretches should be treated with caution, although the main sequencing errors of Illumina platforms are substitution type miscalls (*Kircher, Stenzel & Kelso, 2009*) with the general indel error rate being about an order of magnitude lower (*Laehnemann, Borkhardt & McHardy, 2016*).

Intraspecific chloroplast sequence variability was very low, although the geographical sampling covered almost the entire distribution range and included an accession from

the distant Falkland Islands. The five compared chloroplast genome sequences differed only in 37 variable sites, of which 18 were indels associated with homopolymer stretches and thus of unclear reliability (see preceding paragraph). The remaining variable sites were all autapomorphies, without any phylogenetically informative content. This contrasts with previous studies on intraspecific chloroplast sequence variability in *Jacobaea vulgaris* (32 SNPs observed within 17 individuals, of which 11 were parsimony-informative sites; (*Doorduin et al., 2011*) and *Theobroma cacao* (78 SNPs segregating within 10 individuals; (*Kane et al., 2012*), in which genetic structuring could be observed.

Given the non-existence of genetic structuring in *A. pumila*, it may be speculated that West and Fuegian Patagonia, and the Falkland Islands, have been colonized only recently, probably after the last glacial. Clearly, the sampling in the present study is not adequate to allow for firm conclusions on the Pleistocene history of *A. pumila*, but such a scenario would fit to the classical biogeographic hypothesis brought forward by Villagrán (*1988*; *2001*), based on palynological data: Magellanic moorland species migrated northwards during the last glacial and survived in the lowlands of south-central Chile. From there, they recolonized the large Patagonian Channel region after the disintegration of the Patagonian Ice Sheet, which had reached the continental shelf south of c. 42°S at the height of the last glacial (*Denton & Hughes, 1981*; *Porter, 1981*; *Moreno et al., 2015*). In order to properly and adequately quantify population genetic diversity and identify phylogeographic patterns in *A. pumila*, >350 individuals from almost 40 populations were genotyped at seven nuclear microsatellite loci. These data are currently being analysed and will, together with palaeodistribution modelling, shed light on the open question where *A. pumila* survived the glacials (Pfanzelt et al., 2019, unpublished data).

The phylogenetic reconstruction, based on whole chloroplast genome sequences, recovered Orchidaceae as basally branching within Asparagales. Asteliaceae was then retrieved as sister to the remaining clades of the order (Fig. 2). This topology is in accordance with previous multi-gene-based phylogenetic analyses of Asparagales (*Seberg et al., 2012*). The chloroplast genome of *A. pumila* is the first to be reported for a member of the astelioid clade of basal Asparagales. This is a major improvement in terms of published chloroplast genomes from that order, as especially orchids and subfamily Agavoideae (Asparagaceae) are very well sampled (*McKain et al., 2016*). Furthermore, the generated information—whole genomic DNA shotgun sequences of five *A. pumila* individuals and RNA-Seq data of one of them—represents a valuable genomic resource, e.g., for the identification of nuclear single copy genes. Such markers may prove useful to ascertain the still unresolved infrageneric placement of sect. *Micrastelia*, which contains only *A. pumila* as its single member.

## CONCLUSIONS

The comparison of whole chloroplast genome sequences of five *A. pumila* accessions, sampled from almost the entire distribution range of the species, revealed extremely low levels of sequence variability. The genomic resources generated in the course of the present study may prove useful for future work on *Astelia*, e.g., for the development of single-copy

nuclear markers. These could be employed to ascertain the yet unresolved phylogenetic placement of *A. pumila* within the genus.

## ACKNOWLEDGEMENTS

The authors acknowledge the help of J Fuchs, P Šarhanová and MC García Lino. Centro EULA (University of Concepción, Chile) provided logistic support. Andrew Stanworth and A Davey provided samples from the Falkland Islands (Islas Malvinas). The comments of three anonymous reviewers helped to improve a previous version of the manuscript.

### Funding

This study was part of the research project "The ice-age phylogeography of West Patagonian cushion peat bog vegetation", funded by the German Science Foundation (AL632/7-1, BL462/11), German Academic Exchange Service (D/10/49464) and Dr. Karl Wamsler-Stiftung. The funders had no role in study design, data collection and analysis, decision to publish, or preparation of the manuscript.

### Grant Disclosures

The following grant information was disclosed by the authors:
German Science Foundation: AL632/7-1, BL462/11.
German Academic Exchange Service: D/10/49464.

### Competing Interests

The authors declare there are no competing interests.

### Author Contributions

- Simon Pfanzelt conceived and designed the experiments, performed the experiments, analyzed the data, contributed reagents/materials/analysis tools, prepared figures and/or tables, authored or reviewed drafts of the paper, approved the final draft.
- Dirk C. Albach and K. Bernhard von Hagen conceived and designed the experiments, approved the final draft.

### Field Study Permissions

The following information was supplied relating to field study approvals (i.e., approving body and any reference numbers):

The Chilean Corporación Nacional Forestal and the Falkland Islands Government issued collection permits (Corporación Nacional Forestal: No. 18/2009; Falkland Islands Government: R10/2012.)

### Data Availability

Whole chloroplast genome sequences (accession numbers MH752980–MH752984) and RNA-Seq raw read data of *A. pumila* specimen ALM (SRA accession number SRX4496449) were deposited at NCBI GenBank (https://www.ncbi.nlm.nih.gov/bioproject/483953).

## Supplemental Information

Supplemental information for this article can be found online at http://dx.doi.org/10.7717/peerj.6244#supplemental-information.

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
