# Peer review of "Extremely low levels of chloroplast genome sequence variability in Astelia pumila (Asteliaceae, Asparagales)"

_PeerJ, doi:10.7717/peerj.6244_

## Round 0.1 · original submission · Major Revisions

The main concerns disclosed by the three reviewers coincided in certain issues such as to broaden Introduction by considering more references on the background of the genomics as well as on the study group. In addition the sampling utilized here that comprised only five individuals is not enough to conclude on the genetic structuring. Please change accordingly through the manuscript regarding this subject. Also, consider expanding Discussion in an evolutionary context.

Reviewer 1 ·

Basic reporting

The article is well written and the methods and results are clearly presented. Figures and tables are relevant and clearly present the data and results.

ln 27. The statement that “all major clades” are represented is not informative in terms of ordinal representation. It would be more informative to note the number of Asparagales families or subfamilies that are presented and also to state and provide a citation for family or subfamily recognition e.g. APG III or APG IV. It would also be informative if the families represented in the study were indicated in Figure 2.
ln 40. Three genera are currently recognised in Asteliaceae as per Birch (2015: doi 10.3897/phytokeys.52.4768).
ln 133-134. The wording of the statement “The asteloid clade, including Asteliaceae, is then sister to the remaining Asparagales” implies that other astelioid clade families were represented in this study, which is not the case. To accurately reflect the taxonomic sampling of this study, the results could be stated as “The astelioid clade, represented in this study by Asteliaceae, is then sister to the remaining Asparagales.”
ln 151. Replace “variation associated to homopolymer stretches” with “variation associated with homopolymer stretches”.

Experimental design

The reasoning for the provision of the Astelia pumila chloroplast genome is detailed and the utility of that resource is established. The methods for library preparation and genome assembly are rigorous and are described in sufficient detail so as to be informative for replication of protocols.

ln 159-160. The identification of variable sites as phylogenetically informative or not, should be included in the results.

Validity of the findings

Sampling of Astelia pumila in this study was not sufficient to quantify population genetic diversity and the statement of phylogeographic findings should be modified as noted below.

ln 160. The sampling of this study, including only five individuals, is not sufficient to support genetic structuring conclusions. This statement should be removed or reworded.
ln 163-170. The discussion of how the low levels of sequence divergence documented can be interpreted in a phylogeographic context extrapolates beyond what can be supported by the sampling in this study. The five individuals sampled may or may not accurately quantify the genetic variation present in the species, which is what is required in order to support the conclusion that Astelia pumila has undergone a “post-glacial range expansion” … “out of a single ice-age refugium” (ln 186-187). This discussion is interesting but should be noted as speculative based on the sampling in the current study.

Additional comments

The study represents a significant contribution to genomic resources available for the Asparagales. The authors are commended for an interesting discussion of the evolutionary context of the features of the chloroplast genome that were documented.

Reviewer 2 ·

Basic reporting

The authors miss the goal while trying to address an interesting story on Astelia pumila an enigmatic taxon which is mentioned in the concluding line of the section "conclusion". Here I would appreciate if another reviewer with a strong background in plant taxonomy be invited to review this manuscript to give justice to the scientific authenticity of the concerns raised by the authors.

While going through the manuscript, I expected an interesting story that could raise curiosity behind the experimental setup for sequencing 5 cp genomes of the same species on the pretext of phylogeography.

My major concerns are:
1. Are the plant species collected in different geographical regions?

2. Why the authors have to report 5 different accessions of the same species to look for sequence variability within the species in a plant specimen?

For me it is an apparent waste of time and data redundancy. I would suggest the authors to highlight on the cp DNA of A. pumila and its taxonomic position within Asparagales. The authors rather focus on some highly variable marker regions and sequence them from several populations to arrive at the intended goal. A botanist or a plant taxonomist would be in a better situation to highlight on this.

English usage: Needless to say that the authors have expertise in english. But unfortunately they have written the manuscript hastily or forgot to proofread before uploading the manuscript. they can improvise the ms during the review period.
Some of my observations are as follows:
Abstract: line 16- "via a mapping approach" ...the authors can use the term reference based assembly
Line 32 and line 40 can be synthesized into one single sentence.
Second paragraph in the Introduction section lacks appropriate reference.
Line 46- replace the word "study system" with some appropriate phrase.
Line53-56- Reframe the sentences correctly
Line 72- Expand CONC and OLD
Line 93-96 reframe the sentences with appropriate english grammar usage
Line 99-100 Please reframe the sentences in such a way that the tool web urls are placed immediately after they are mentioned in the text.

Experimental design

Research question is not well defined. I don't find any interesting knowledge gap either in the introduction nor in the discussion part.

The authors need to provide an interesting preamble to the research question and their work towards resolving it.

The authors also need to categorize the methods section into sub sections as detailed below:

sample collection, DNA extraction and sequencing validation
Data processing
Genome annotation
Phylogeny reconstruction
etc.....

I would suggest the authors to go for SSR analysis and phylogeny based on SNP identification (using KSNP3) and identify potential RNA editing sites.

Validity of the findings

Whatever the authors have presented in the manuscript in the results section are error free. But I have a strong reservation regarding the validity of the approach to the research question by the authors.

Following are my observations:

As long as the manuscript deals with reporting of the cp DNA of A. pumila I am ok with the findings.
But as per the title of the manuscript the authors have to collect more samples, identify some interesting markers and then perform network analysis.
I would suggest the authors to follow this particular manuscript while revising their manuscript and redesigning their experiments:
"Comparative systematics and phylogeography of Quercus Section Cerris in western Eurasia: inferences from plastid and nuclear DNA variation"
https://peerj.com/articles/5793/?utm_source=summary_email_publication&utm_medium=email&utm_campaign=connection

Additional comments

I request the authors to use the cp genomes generated by them in generating a relevant and exciting manuscript. They should follow the manuscript titled "Comparative systematics and phylogeography of Quercus Section Cerris in western Eurasia: inferences from plastid and nuclear DNA variation" in getting better ideas for developing their own paper.

·

Basic reporting

1. Introduction_ Line 39 _ need Reference.
2. On Introduction_ Line 40. Repeated “belongs to Asteliaceae” as mentioned at the first line of Introduction.
3. Introduction _Line 50 _After “ did not provide satisfactory results.” _need references here.
4. Material Methods_ Line 78_ Geneious program need reference.
5. In Material methods _ to check the junctions among LSc, SSc and IR you used Sanger Sequencing, why? As you have hundreds or thousands of these sequences reads within your genomic sequence. You just need to separate these regions (junction regions with some more nucleotides before and after) then make them as a reference, next map them to whole genomic sequences to see the error, if present, I don’t think that the only one sequence better than the whole genomic.
6. Line 95, what was the reference genome that you used as the source of annotation chloroplast genomes?
7. Line 104, the citation (Huson & Bryant 2006) is (2005) in the reference list.
8. Line 106, citation (Sensu Chase, Reveal, & Fay, 2002) the year is 2009 in the reference list.
9. Line 120 _ the end of the sentence_ after “variation was very low in A. pumila” you can cite some reference like “Wittzell, 1999; Särkinen 2013; and Salih et al. 2017”
10. Line 154_ Laehnemann, Borkhardt & McHardy, 2015)_ the year is 2016 in the Reference list.
11. Line 155_ after “variability was very low” cite some reference such as “Wittzell, 1999; Särkinen 2013; and Salih et al. 2017”.

Experimental design

no comment

Validity of the findings

no comment

Additional comments

1. Plastome genome assembly is not easy step to get complete circular sequence using Illumina short sequences. Sometime there are many assembly errors are also happened that was reported in Scientific Reports 5: 15655. However, authors do not explain how they obtained the complete circular sequence and did not cite any other protocol.
2. Data is not descriptive enough, though summarized. As in Introduction there is just only 4 references cited.
3. Would be good to cite other recent papers comparing chloroplast genomes in several closely related taxa, such as “Wittzell, 1999; Särkinen 2013; and Salih et al. 2017”.
4. What is your suggestion about choosing chloroplast marker (single gene or spacer) for the family future study? Is there any recommended region to use to get good results as whole chloroplast genome?
5. You did the whole genomic sequences of these plants, as it should be there is whole chloroplast, nuclear, and some mitochondrial sequences. Have you assembled the whole rDNA sequences? So as to check the level of variability on ribosomal DNA.
6. As you are the first to sequence the whole chloroplast genome of Asteliaceae, it is better to check the inventions in chloroplast genome to be as the reference for other chloroplast genome for this family in the future, there is a large inversion in chloroplast which is located within the LSC region and another one is the whole SSC inversion.

---

## Round 0.2 · Minor Revisions

I appreciate your effort considering suggestions by the reviewers. However in my opinion the Introduction is still incomplete, you focused in providing information on Astelia pumila, but my suggestion is to include previous research on chloroplast genomes in Asparagales and in other monocots. In way to have a perspective to understand better how this genome varies in these groups and what to expect in this species. In addition read carefully the paper again, I found some typos.

---

## Round 0.3 · accepted · Accept

I appreciate the changes made in this version, Introduction and Discussion were most improved.

#